# The Potential Benefits of a Novel Food Supplement Based on Cannabis Sativa, Boswellia, and Fish Oil for Pain and Inflammation in Physical Activity: Unraveling the Role of Orexin-A Modulation

**DOI:** 10.3390/sports13070199

**Published:** 2025-06-22

**Authors:** Antonietta Messina, Antonietta Monda, Valentina Vassallo, Girolamo Di Maio, Rita Polito, Marco La Marra, Salvatore Allocca, Maria Casillo, Fiorenzo Moscatelli, Cristina Scavone, Francesco Taturi, Vincenzo Monda, Giovanni Messina, Chiara Schiraldi, Marcellino Monda

**Affiliations:** 1Department of Precision Medicine, University of Campania “Luigi Vanvitelli”, 80138 Naples, Italy; antonietta.messina@unicampania.it; 2Department of Human Science and Quality of Life Promotion, San Raffaele Telematic University, 00166 Rome, Italy; antonietta.monda@uniroma5.it; 3Department of Experimental Medicine, Section of Human Physiology, Unit of Dietetics and Sports Medicine, University of Campania “Luigi Vanvitelli”, 80138 Naples, Italy; valentina.vassallo@unicampania.it (V.V.); salvatore.allocca@unicampania.it (S.A.); maria.casillo@unicampania.it (M.C.); chiara.schiraldi@unicampania.it (C.S.); marcellino.monda@unicampania.it (M.M.); 4Department of Life Science, Health, and Health Professions, Link Campus University, 00165 Rome, Italy; cristina.scavone@unilink.it; 5Department of Psychology and Health Sciences, Pegaso Telematic University, 80143 Naples, Italy; girolamo.dimaio@unipegaso.it (G.D.M.); rita.polito@unipegaso.it (R.P.); 6Department of Education and Sport Sciences, Pegaso Telematic University, 80143 Naples, Italy; fiorenzo.moscatelli@unipegaso.it; 7Heracle Lab Research in Educational Neuroscience, Niccolò Cusano University, 00166 Rome, Italy; francesco.tafuri@unicusano.it; 8Department of Economics, Law, Cybersecurity, and Sports Sciences, University of Naples “Parthenope”, 80133 Naples, Italy

**Keywords:** Cannabis sativa, Boswellia serrata, food supplements, inflammation, joint pain, orexin-A, metabolic syndrome, physical activity

## Abstract

Background: Nutraceuticals have emerged as promising alternatives to conventional pharmacological treatments for managing joint pain and low-grade inflammation in physically active individuals. However, few clinical studies have evaluated the combined metabolic, inflammatory, and neuroendocrine effects of multi-ingredient supplements. This study aimed to evaluate the effects of Flector Softgel FS Integratore, a multi-component food supplement, on joint pain, inflammatory markers, metabolic health, and orexin-A levels in physically active adults. Methods: In this randomized, controlled, low-intervention study, 25 adult participants (aged 30–60 years and amateur athletes engaging in at least 3 sessions/week of moderate physical activity) were assigned to either a treatment group (n = 15 received Flector Softgel FS for 14 days) or a placebo group (n = 10). The supplement contained 500 mg of Cannabis sativa seed oil (THC-free), 250 mg of Boswellia serrata extract, 250 mg of fish oil, 160 mg of omega-3 fatty acids, and 0.6 mg of undenatured type II collagen (UC-II). Pain was assessed using the Visual Analog Scale (VAS). Metabolic parameters, inflammatory cytokines (IL-6, IL-8, TNF-α, IFN-γ, and IL-10), and serum orexin-A levels were measured before and after the intervention. Results: Compared with the placebo, the treatment group showed a significant reduction in VAS scores (*p* < 0.001), as well as improvements in BMI, insulin, and lipid profiles, and a decrease in pro-inflammatory cytokines (IL-6, IL-8, TNF-α, and IFN-γ). A reduction in orexin-A levels was also observed in the treatment group (*p* < 0.001), with a positive correlation between orexin-A and perceived pain. No adverse effects were reported. Conclusions: Flector Softgel FS Integratore may be effective in reducing joint pain and systemic inflammation while supporting metabolic health in active adults. These effects may involve indirect modulation of orexin-A, though the exact mechanisms remain to be clarified. Despite the promising results, conclusions regarding efficacy in comparison with NSAIDs should be approached with caution in the absence of a pharmacological control group. Further studies with larger samples and a longer duration are needed.

## 1. Introduction

Engaging in regular physical activity is fundamental to overall health and well-being and plays a key role in preventing various chronic conditions, particularly metabolic and cardiovascular diseases [1]. However, physical activity is not without risks; it may predispose individuals to injuries and musculoskeletal overload syndromes, especially in athletes and highly active individuals [2]. Musculoskeletal disorders remain a leading cause of chronic disability worldwide, posing a considerable burden on healthcare systems [3]. Conventional pharmacological treatments typically involve the use of analgesics or anti-inflammatory agents (e.g., acetaminophen or NSAIDs), which, although effective, are often associated with adverse effects, such as gastrointestinal distress, bleeding, constipation, and cognitive impairment [4]. In recent years, interest in supplements and nutraceuticals as alternatives or adjuncts to traditional pharmacological therapies has grown significantly. These compounds aim to alleviate pain and inflammation while minimizing the risk of systemic side effects, especially in chronic users. Several natural compounds have demonstrated promising anti-inflammatory and analgesic properties, sometimes comparable to pharmacological agents [5,6,7]. Among the most studied is Cannabis sativa, a plant with a complex chemical profile, whose cannabinoids—particularly cannabidiol (CBD)—exhibit potent anti-inflammatory and analgesic effects. Unlike THC, CBD is non-psychoactive and permitted in many European countries, including Italy. CBD exerts its effects by modulating the endocannabinoid system (ECS), inhibiting the production of pro-inflammatory cytokines, and regulating key receptors, such as TRPV1 and PPARγ [8,9,10,11]. Cannabis has a long history of use as an analgesic, with modern studies confirming its efficacy in managing inflammatory pain [12]. Another valuable nutraceutical agent is Boswellia serrata, whose bioactive component, boswellic acid, is a 5-lipoxygenase inhibitor known to reduce leukotriene-mediated inflammation in musculoskeletal and joint conditions [13,14,15]. In parallel, neuroendocrine factors have emerged as promising targets for pain modulation. One such molecule, orexin-A (hypocretin-1), synthesized in the lateral hypothalamus, has been shown to influence pain perception, energy balance, sleep, and metabolic rate. It acts via OX1R and OX2R receptors, modulating both nociceptive pathways and the release of key neurotransmitters such as GABA and glutamate [16,17,18,19,20]. Furthermore, orexin-A may interact with the endogenous opioid system and exert anti-inflammatory effects, thereby indirectly contributing to pain relief [21,22,23]. The interaction between physical activity, pain perception, and quality of life has been extensively documented. In particular, Puszczalowska-Lizis et al. (2020) observed that structured physical activity not only improves physical function but also significantly enhances psychosocial well-being and overall quality of life, even in clinical populations, such as women post-mastectomy [24]. This suggests that interventions capable of reducing exercise-related musculoskeletal pain can have far-reaching impacts on functional capacity and life quality, especially in mid-life individuals striving to maintain regular physical activity.

Flector Softgel FS Integratore is a multi-component nutraceutical formulation developed to support joint function and reduce inflammation, particularly in individuals experiencing musculoskeletal discomfort associated with physical activity. The formulation acts via multiple complementary pathways, including inflammatory cascade inhibition via 5-LOX inhibition (Boswellia), SPM production (omega-3), and the suppression of pro-inflammatory cytokines (Cannabis sativa seed oil and fish oil). In addition, it protects joint tissue through immune tolerance induced by UC-II and the antioxidant properties of omega-3 fatty acids, and potentially, it is mediated by reduced systemic inflammation, which may influence orexin-A regulation, appetite, and energy homeostasis. The combination of these ingredients is designed to exert synergistic effects: reducing joint pain, suppressing inflammation, improving mobility, and potentially supporting metabolic balance in active individuals experiencing low-grade inflammation.

To address the exploratory nature of previous approaches, this study is grounded in the following pre-specified hypotheses: (I) the oral administration of the supplement Flector Softgel FS Integratore, containing Cannabis sativa extract (THC-free), reduces exercise-induced musculoskeletal pain in active adults compared with baseline; (II) the supplement also improves joint inflammatory status and recovery, potentially enhancing training continuity and perceived quality of life.

For these reasons, the primary objective of this study was to evaluate the potential benefits of a novel food supplement containing Cannabis sativa extract (THC-free), Boswellia serrata, and fish oil for pain perception and inflammatory responses associated with physical activity in active adults. Special attention was given to the modulatory role of orexin-A (hypocretin-1), a neuropeptide involved in pain regulation, inflammation, and energy homeostasis. In addition, this study aimed to assess the efficacy of the supplement in reducing exercise-induced musculoskeletal pain and investigate changes in circulating levels of orexin-A and inflammatory cytokines (e.g., IL-6 and IL-10) as possible underlying mechanisms.

## 2. Materials and Methods

### 2.1. Participants and Study Protocol

This investigation was designed as a pilot study, conducted with the aim of exploring the potential anti-inflammatory and analgesic effects of Flector Softgel FS Integratore in a small sample of physically active subjects. At the time this study was carried out, the product was not yet commercially available, and the formulation was provided exclusively for research purposes.

This was a randomized, controlled, low-intervention, parallel-group trial, designed to evaluate the effects of a food supplement on pain and inflammation in physically active adults. This study was conducted according to the Declaration of Helsinki and approved by the Ethics Committee of the University of Campania “Luigi Vanvitelli” (Prot. 35439/2022, issued on 30 November 2022). A total of 25 adult participants (aged 30–60 years; both sexes) were recruited from the UOC of Dietetics, Sports Medicine, and Psycho-Physical Well-Being–AOU “L. Vanvitelli”, Naples, Italy, through local advertisements and screening at outpatient clinics. The participants were randomized into two groups according to a computer-generated list:-Treatment group: Fifteen participants received Flector Softgel FS Integratore for 2 weeks.-Placebo group: Ten participants received a placebo supplement identical in appearance and taste.

A physician and a research assistant not involved in outcome evaluation handled subject allocation and data collection to reduce bias.

The sample size was estimated based on a power analysis (α = 0.05; β = 0.20) to detect a minimum difference of 20 mm in the VAS pain scores between groups, with a standard deviation of 15 mm, assuming a two-tailed test. This yielded a required sample size of 12 subjects per group. Considering potential dropouts, 25 participants were enrolled.

### 2.2. Physical Activity Program

Participants were defined as amateur physically active individuals in accordance with the WHO recommendations, which suggest a minimum of 150 min per week of moderate-intensity aerobic activity for adults. Specifically, all included subjects reported engaging in at least 150 min of brisk walking per week, distributed over no fewer than three sessions. This type and amount of activity corresponds to a moderate-intensity level and represents a common form of physical exercise among non-athlete populations. The choice to include individuals performing regular walking was made to reflect a real-world, generalizable population of active adults who may experience mild joint discomfort or low-grade inflammation, and who could benefit from non-pharmacological interventions.

### 2.3. Inclusion and Exclusion Criteria

The inclusion criteria are reported as follows:-Male and female participants aged 30–60 years.-Amateur athletes with regular physical activity (>3 times/week).-A body mass index (BMI) between 22 and 30 kg/m^2^ (normal–overweight range).-A history of recurrent joint or musculoskeletal pain during or after physical activity.-No anti-inflammatory or joint-active medication use in the 10 days prior to enrollment.-A negative pregnancy test for women of childbearing potential.-Provision of written informed consent.

Exclusion criteria are reported as follows:-A diagnosed chronic inflammatory or autoimmune disease.-The use of corticosteroids, opioids, or immunomodulators within 30 days.-Allergies to any supplement ingredients.-Pregnancy or lactation.-Psychiatric disorders or cognitive impairments affecting compliance.

At baseline, the following sociodemographic variables were collected via a structured interview: age, sex, educational level, employment, and physical activity frequency. The clinical variables included BMI, waist circumference, and reported pain history.

### 2.4. Flector Softgel FS Integratore Characteristics

According to the instructions on the label, Flector Softgel FS Integratore is intended exclusively for oral use.

Flector Softgel FS Integratore (1 cps) contains the following:An amount of 500 mg of Cannabis Sativa L. seed oil (cannabinoid-free) (recognized by the Ministry of Health for improving joint functionality);An amount of 250 mg of Boswellia Serrata Roxb (supports joint functionality and helps counteract localized tension states);An amount of 250 mg of fish oil 33/23 Tg (related to anti-inflammatory activity);An amount of 160 mg of omega-3;An amount of 0.6 mg of collagen UC II (undenatured type II; supported by clinical studies).

### 2.5. Anthropometrical and Biochemical Measurements

Height, weight, BMI, and waist circumference were measured using standardized procedures (SECA 700 scale ± 0.1 kg). The blood analysis included the glucose, insulin, lipid profile, and C-reactive protein assay. In addition, orexin-A serum levels were evaluated by the ELISA test, as reported in a data sheet (Elabscience, Houston, TX, USA). All ELISA samples were run in triplicate.

### 2.6. Pain Evaluation

Two participant groups were subjected to the Visual Analogue Scale (VAS) to evaluate the magnitude of pain. Indeed, the VAS is the visual representation of the magnitude of pain the subject feels. It is represented by a 10 cm long line in the original validated version, with or without markings at each centimeter. One end indicates no pain and corresponds to 0, while the other end indicates the worst imaginable pain and corresponds to 10. The scale is manually completed by the patient, who is asked to mark a point on the line that represents their perceived pain. The distance measured from the 0 end corresponds to the subjective measure of pain. The patient marks the point on the line that depicts the intensity of their pain at that moment; the examiner scores it using a graduated scale on the back of the ruler. The scale line can be oriented horizontally or vertically without affecting its sensitivity [9], although some studies have shown that the horizontal version has a lower failure rate [10]. In other versions, the scale can be administered using a laminated support with a movable slider to indicate the perceived pain level. As a unidimensional measure, the scale is used to evaluate current pain or pain experienced in the last 24 h [9]. The score is calculated in mm by measuring with a ruler the length of the line from the end corresponding to the minimum intensity to the mark made by the patient. Participants marked their perceived pain intensity; scores were categorized as follows:0–4 mm: No pain;5–44 mm: Mild pain;45–74 mm: Moderate pain;75–100 mm: Severe pain.

The VAS was administered at baseline and post-treatment to both groups. The VAS represents an ordinal measure as it has the property of expressing values in a magnitude order, comparable to each other. The scale aims to measure a characteristic or attitude believed to extend over a continuum of values, which cannot be easily measured directly [9,10,11,12].

### 2.7. Bioplex Assay

To assess the modulation of specific soluble pro- and/or anti-inflammatory mediators, an 8-plex immunoassay panel (Bio-Rad Laboratories s.r.l., Milan, Italy) based on xMAP technology was employed. This panel contained the following biomarkers: IL-2, IL-4, IL-6, IL-8, IL-10, INF-γ, TNF-α, and GM-CSF. The experimental assay was performed by following a previously described protocol [24]. Briefly, before (T0) and after the treatments (T1), the serum of two patient groups (FS, food supplement; P, placebo) was stored at −80 °C until the analyses. In agreement with the manufacturer’s instructions and using a Bio-Plex array reader (Luminex, Austin, TX, USA), the samples were diluted 1:4, and the assay procedures were carried out. Specific analyte concentrations (pg/mL) were assessed by a standard curve according to the constructor’s protocol. In the Results Section, we report only data about the biological mediators significantly (*p* < 0.05) modulated with respect to other patient groups or treatment times. Each sample was analyzed in triplicate, and the data are shown as means ± SDs.

### 2.8. Description of Study Variables

In this study, the main independent variable was the type of treatment administered, specifically the intake of Flector Softgel FS Integratore compared with a placebo. Participants were randomly assigned to one of the two groups, and this allocation formed the basis for evaluating the treatment’s effects on the clinical and biochemical parameters under investigation. The primary dependent variables were selected to capture the potential clinical impact of the intervention on pain perception and inflammatory response. These included the following:-Subjective pain perception, measured using the Visual Analogue Scale (VAS);-The serum orexin-A concentration;-The levels of the pro-inflammatory cytokine IL-6.

These outcomes were chosen for their relevance in reflecting both inflammatory activity and the neurophysiological modulation of pain in physically active individuals.

As secondary outcomes, this study included the following:-Serum levels of the anti-inflammatory cytokine IL-10 and self-reported measures of post-exercise recovery and perceived well-being, obtained through structured questionnaires.

To ensure the validity of the findings, particular attention was given to identifying and accounting for potential confounding variables, factors that could influence both the exposure (treatment) and the outcomes.

These included baseline pain levels (the VAS at T0), age and sex, body mass index (BMI), and the participants’ habitual physical activity level.

These variables were collected during the initial assessment and, through randomization, were expected to be evenly distributed between the treatment groups. This structured approach allowed for a targeted assessment of the supplement’s efficacy while controlling for external influences and focusing on the biological mechanisms underlying pain and inflammation modulation in an athletic population.

### 2.9. Statistical Analysis

Statistical analyses were conducted using the StatView software, version 5.0.1.0 (SAS Institute Inc., Cary, NC, USA). Prior to hypothesis testing, the normality of all continuous variables was evaluated using the Shapiro–Wilk test. Variables that followed a normal distribution were analyzed using parametric tests, including unpaired *t*-tests for between-group comparisons and paired *t*-tests for within-group comparisons (pre- vs. post-treatment). For variables that did not meet the assumption of normality, non-parametric tests were employed: the Mann–Whitney U test for between-group comparisons and the Wilcoxon signed-rank test for within-group changes. A two-tailed *p*-value ≤ 0.05 was considered statistically significant. Quantitative data are presented as means ± standard deviations (SDs) for normally distributed variables, and as medians with interquartile ranges (IQRs) for non-normally distributed variables, as appropriate. In this study, the independent variable was the intervention group (Flector Softgel FS Integratore vs. placebo). The primary dependent variables included the Visual Analogue Scale (VAS) score for pain, serum levels of orexin-A, and IL-6 concentrations. The secondary dependent variables included IL-10 levels, metabolic and inflammatory biomarkers (e.g., CRP, insulin, and lipid profile), and subjective recovery indicators. To account for potential confounders, the baseline demographic and clinical characteristics (age, sex, BMI, and habitual physical activity level) were collected and balanced between groups via randomization. This statistical approach was designed to rigorously evaluate the effects of the supplement on inflammation, pain modulation, and metabolic responses in an active adult population.

## 3. The Cytokines Assay in Flector Softgel FS Group and Placebo Group

### 3.1. Anthropometric and Biochemical Parameters

The descriptive and comparative statistics of anthropometric and biochemical parameters for both the treatment and placebo groups are presented in Table 1. All variables were tested for normality using the Shapiro–Wilk test, and the results are provided in Table 2. Variables with a normal distribution were analyzed using parametric tests and expressed as means ± standard deviations (SD); non-normally distributed data were presented as medians [interquartile ranges] and analyzed using non-parametric methods.

In the Flector Softgel FS group, a significant reduction in body weight (*p* = 0.043) and BMI (*p* = 0.038) was observed after 2 weeks. Improvements were also noted in lipid profile, with significant reductions in LDL cholesterol (*p* = 0.015) and triglycerides (*p* = 0.009), along with a decrease in insulinemia (*p* = 0.021) and C-reactive protein (CRP) (*p* = 0.012). No significant changes were observed in the placebo group for any parameter.

### 3.2. Orexin-A and VAS Profile in Flector Softgel FS Group and Placebo Group

As shown in Figure 1a, a significant reduction in orexin-A levels was observed in the treatment group compared with the placebo group following the intervention [F(1.508, 23.12) = 16.62; *p* < 0.001]. Similarly, the VAS pain scores showed a substantial decrease in the treatment group [F(1.281, 15.38) = 18.37; *p* < 0.001] (Figure 1b).

In addition, a positive correlation was found between orexin-A levels and VAS scores at T1 (Figure 2), suggesting a potential relationship between this neuropeptide and pain perception [F = 4.54; R^2^ = 0.16; *p* = 0.041].

### 3.3. Bioplex Assay

The bioplex assay results are shown in Figure 3. In detail, it is possible to note that among the soluble mediators analyzed, IL-6, IL-8, TNF-α, and INF-γ resulted as being significantly (*p* < 0.05) modulated in the patients’ serum. Specifically, the following graphs displayed that FST1 presented a significant (*p* < 0.05) reduction with respect to FST0 for each reported analyte. Among the P group, there was a slight increase in IL-6, IL-8, and INF-γ between time 0 and after 15 days, while the level of TNF-α remained very similar (Figure 3c).

## 4. Discussion

The findings of this pilot study provide preliminary evidence supporting the potential efficacy of Flector Softgel FS, a novel nutraceutical formulation, in reducing exercise-related joint pain, modulating systemic inflammation, and improving metabolic parameters in physically active adults. The primary endpoint, pain reduction, was accomplished with a statistically and clinically significant decrease in VAS scores after two weeks of supplementation. This suggests that the formulation may offer a non-pharmacological option for pain relief in individuals performing low- to moderate-intensity physical activity. However, while these results are promising, direct comparisons with conventional medications, such as NSAIDs, must be interpreted with caution, as this study did not include a pharmacological (positive) control group. An objective comparison with NSAIDs would require a dedicated trial design with appropriate dosing and endpoints.

Beyond pain relief, Flector Softgel FS demonstrated a notable capacity to modulate inflammatory activity, with significant reductions in IL-6, IL-8, TNF-α, and IFN-γ. These changes were not observed in the placebo group, where pro-inflammatory markers showed mild increases, reinforcing the supplement’s potential anti-inflammatory role. These results are consistent with the known mechanisms of action of its active components: Cannabis sativa seed oil (free of THC), Boswellia serrata, and fish oil, all recognized for their immunomodulatory properties. The supplement was also associated with improvements in BMI and insulin sensitivity, suggesting systemic metabolic benefits [25]. These findings align with the previous literature linking low-grade inflammation to metabolic dysregulation and support the idea that modulating inflammation can aid in weight management and metabolic health [26,27]. A notable and novel finding of this study was the observed reduction in circulating orexin-A levels in the treatment group following supplementation. Orexin-A (hypocretin-1) is a neuropeptide produced by the lateral hypothalamus, classically involved in arousal, appetite regulation, energy homeostasis, and more recently, pain modulation. While the precise mechanisms linking orexin-A to peripheral pain and inflammation are not yet fully elucidated, existing research suggests that elevated orexin-A levels may contribute to enhanced sympathetic activity and amplified nociceptive signaling, particularly under inflammatory conditions [28]. Orexin-A receptors (OX1R and OX2R) are expressed in several regions involved in pain processing, including the periaqueductal gray, the dorsal horn of the spinal cord, and the thalamus. Activation of these receptors has been associated with both pro- and anti-nociceptive effects, depending on the context and route of modulation. The reduction in orexin-A levels observed in this study may reflect a downstream consequence of improved systemic inflammatory status, as supported by the concurrent decreases in IL-6, TNF-α, and other pro-inflammatory cytokines [29]. Given that inflammation itself can influence hypothalamic neuropeptide signaling, it is plausible that the anti-inflammatory action of the supplement indirectly modulated orexin-A secretion, reducing sympathetic tone and nociceptive sensitivity [30]. Alternatively, orexin-A suppression may be related to improved metabolic balance, as this neuropeptide is also implicated in feeding behavior and insulin sensitivity [31].

However, due to the exploratory nature of this analysis and the absence of central nervous system measurements or mechanistic assays, these interpretations must be viewed as hypothesis-generating rather than conclusive. Future studies should incorporate neuroimaging, cerebrospinal fluid markers, or central receptor profiling to better understand how nutraceutical interventions may interact with the orexinergic system in the context of pain and inflammation regulation [32,33]. Recent research strongly supports the use of nutraceuticals for inflammation and exercise recovery. For instance, Perez-Pinero et al. (2023) showed that omega-3 and Boswellia supplementation significantly improved joint stiffness and reduced soreness in amateur runners [34]. Similarly, Marx et al. (2021) reported that Cannabis sativa-derived cannabidiol (CBD) can reduce pain perception and inflammation after resistance training, with effects linked to TRPV1 and PPARγ pathways, mechanisms also relevant to components in our formulation [35]. Additionally, Pawar et al. (2023) emphasized the value of multi-component nutraceuticals in modulating cytokine cascades and improving lipid metabolism, confirming the rationale behind combining anti-inflammatory and metabolic ingredients, as we did in this study [36]. Unlike these previous studies, which primarily focused on isolated compounds or subjective outcomes, our work offers a more integrated approach, combining clinical, biochemical, and neuroendocrine markers to assess the effect of a multi-target supplement in a real-world active population [8,37,38]. This approach allows for a more nuanced understanding of how nutraceuticals may influence systemic inflammation, pain, and metabolism in parallel. This study contributes to the current literature by being one of the first to evaluate a comprehensive multi-compound nutraceutical for joint pain and inflammation in the context of physical activity, while also exploring orexin-A as a potential mediator [39,40,41]. The combined analysis of subjective (VAS), biochemical (cytokines), and metabolic parameters (BMI, insulin, and lipids) represents a novel, multidimensional model for assessing the efficacy of nutritional interventions. These findings support the potential for non-pharmacological approaches to managing low-grade inflammation, joint discomfort, and related metabolic stress in active adults [42,43]. This study has several limitations. The sample size was relatively small, which limits the generalizability of the findings. Although a power analysis supported the number of participants, larger cohorts are needed to confirm the results. Second, the intervention duration was short (2 weeks), limiting the ability to assess sustained or long-term effects. Third, the analysis of orexin-A was exploratory, and this study was not specifically designed to elucidate its mechanistic role. Additionally, dietary intake and physical activity intensity were not controlled during the study period, which could act as confounding factors.

Finally, while the randomization procedure helped to minimize selection bias, blinding effectiveness was not formally tested, and subjective outcomes like VAS may be influenced by expectation effects.

## 5. Conclusions

This study provides preliminary evidence that Flector Softgel FS, a nutraceutical supplement containing Cannabis sativa seed oil (THC-free), Boswellia serrata, and fish oil, may offer significant benefits for pain management and inflammation control in physically active adults experiencing joint discomfort. The supplement was associated with a reduction in subjective pain (VAS scores), a favorable modulation of key inflammatory markers (IL-6, IL-8, TNF-α, and IFN-γ), and improvements in metabolic parameters such as BMI, insulin, and lipid profiles. Notably, this study also explored the potential link between supplementation and orexin-A modulation, which may reflect broader interactions between inflammation, metabolism, and neuropeptide regulation, although this remains speculative and warrants further investigation. These findings suggest that Flector Softgel FS could represent a safe, well-tolerated, and multi-targeted approach for managing low-grade inflammation, exercise-induced joint pain, and metabolic imbalances, particularly in populations seeking non-pharmacological alternatives to NSAIDs. However, due to the limited sample size and short duration of the intervention, further large-scale, long-term, and mechanistic studies are necessary to validate these outcomes and better understand the underlying pathways involved. In addition, this was a pilot study, conducted with the aim of exploring the potential anti-inflammatory and analgesic effects of Flector Softgel FS Integratore in a small sample of physically active subjects. At the time this study was carried out, the product was not yet commercially available, and the formulation was provided exclusively for research purposes. The results presented here should be considered preliminary and serve as a foundation for future studies involving larger populations, longer intervention periods, and the inclusion of pharmacological control groups.

## Figures and Tables

**Figure 1 sports-13-00199-f001:**
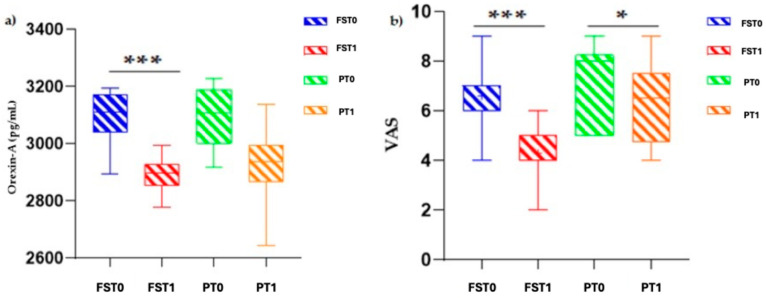
Effects of treatment on orexin-A levels and VAS scores. (**a**) reports orexin-A in treated group at T0 (FST0) and at T1 (FST1) compared with placebo group at T0 (PT0) and T1 (PT1); (**b**) reports VAS in treated group at T0 (FST0) and at T1 (FST1) compared with placebo group at T0 (PT0) and at T1 (PT1). *** *p* < 0.001; * *p* < 0.05.

**Figure 2 sports-13-00199-f002:**
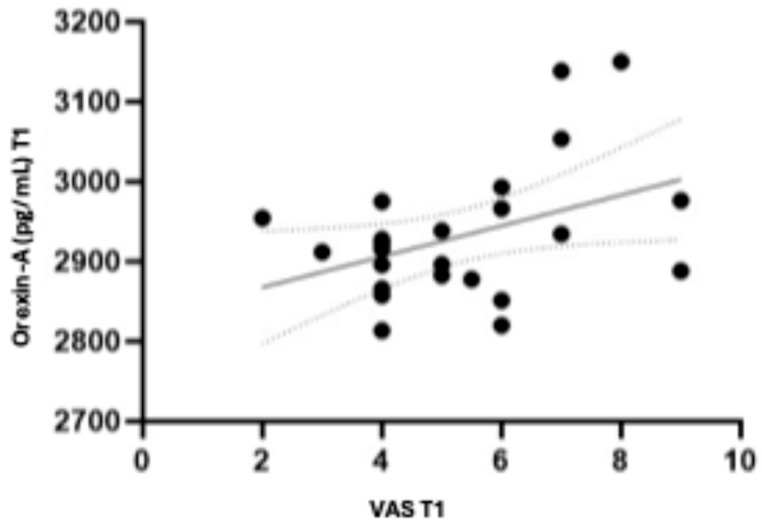
Correlation Between orexin-A and VAS scores at T1. In Figure 2, the overall linear regression between orexin-A and VAS at T1 is reported; in particular, there is a positive correlation between orexin-A and VAS.

**Figure 3 sports-13-00199-f003:**
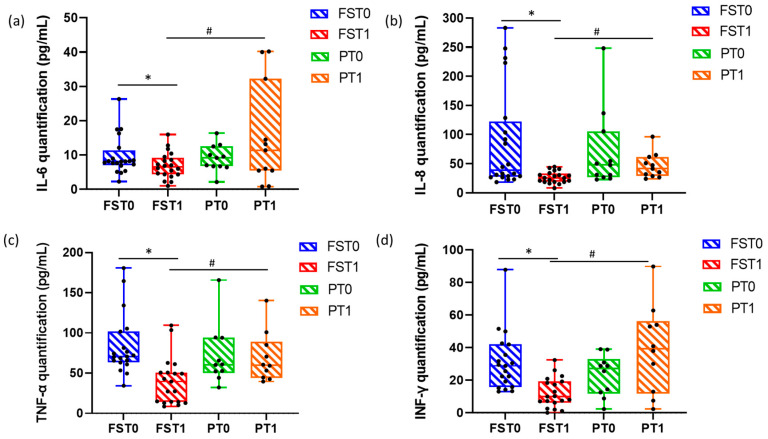
Serum cytokine modulation in treatment and placebo groups. IL-6 (**a**), IL-8 (**b**), TNF-α (**c**), and INF-γ (**d**) resulted as being significantly modulated in the patients’ serum. Multiplex assay. Test was performed to assess statistical differences between data. * *p* < 0.05 at T1 vs. T0; # *p* < 0.05 in PT1 vs. FST1.

**Table 1 sports-13-00199-t001:** Anthropometric and biochemical parameters before and after treatment.

Parameter	Flector FS T0	Flector FS T1	*p*-Value (Flector)	Placebo T0	Placebo T1	*p*-Value (Placebo)
Weight (kg)	74.46 ± 2.97	73.65 ± 2.98	0.043	75.35 ± 2.98	75.55 ± 2.94	n.s.
BMI (kg/m^2^)	24.46 ± 0.63	23.19 ± 0.62	0.038	25.64 ± 0.83	25.72 ± 0.87	n.s.
Total cholesterol (mg/dL)	150.13 ± 5.77	140.91 ± 2.93	0.027	139.45 ± 3.50	137.32 ± 2.40	n.s.
LDL (mg/dL)	111.32 ± 6.48	90.57 ± 5.72	0.015	99.45 ± 1.20	98.56 ± 2.30	n.s.
Triglycerides (mg/dL)	125.14 ± 5.27	73.25 ± 6.14	0.009	110.45 ± 2.60	111.12 ± 3.50	n.s.
HDL (mg/dL)	55.13 ± 6.14	53.76 ± 4.14	n.s.	48.24 ± 2.50	49.45 ± 4.60	n.s.
Insulinemia (µU/mL)	5.53 [4.10–6.75]	3.37 [2.10–4.25]	0.021	5.40 ± 3.10	4.67 ± 1.30	n.s.
CRP (mg/mL)	0.89 ± 0.10	0.48 ± 0.07	0.012	0.78 ± 0.50	0.77 ± 0.20	n.s.

Anthropometric and biochemical parameters before (T0) and after (T1) treatment in the Flector Softgel FS and placebo groups. Data are presented as means ± standard deviations (SDs) for normally distributed variables, and as medians [interquartile ranges] for non-normally distributed variables. Statistical comparisons were performed within each group (T1 vs. T0) using paired *t*-tests or Wilcoxon signed-rank tests, based on the Shapiro–Wilk normality test results. *p*-values refer to within-group comparisons. BMI—body mass index; LDL—low-density lipoprotein; HDL—high-density lipoprotein; CRP—C-reactive protein; n.s.—not significant (*p* > 0.05).

**Table 2 sports-13-00199-t002:** Shapiro–Wilk normality test for all variables.

Variable	W Statistic	*p*-Value	Distribution
Weight	0.974	0.652	Normal
BMI	0.979	0.734	Normal
Total cholesterol	0.966	0.544	Normal
HDL	0.912	0.043	Non-normal
LDL	0.983	0.782	Normal
Triglycerides	0.961	0.478	Normal
Insulinemia	0.894	0.032	Non-normal
CRP	0.927	0.060	Normal

## Data Availability

All data are available upon request from the corresponding author.

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
