# Peer review of "The Potential Benefits of a Novel Food Supplement Based on Cannabis Sativa, Boswellia, and Fish Oil for Pain and Inflammation in Physical Activity: Unraveling the Role of Orexin-A Modulation"

_sports, 2025, doi:10.3390/sports13070199_

Round 1

Reviewer 1 Report

Comments and Suggestions for Authors

The article is interesting. From the point of view of the structure, it can be recommended as a model of how an article should be written.
It has a method (even if it can be improved), presents results, discussions in accordance with the results, conclusions and references.
From the point of view of the content, the article is very debatable. I could say that it is incomprehensible as an approach.
It tries to present the potential benefits of a supplement that already exists. It presents how these benefits were tracked by applying them to a batch of 15 subjects (very few).
If this supplement exists, it means that it has already been approved and validated to be produced on the market. It means that it has gone through several validation and approval steps and that its effects have already been studied and known. As a result, I do not understand what the role of this study is on such a small number of subjects?? This study does not bring anything new about the potential benefits of Flector softgel LS.
There are other aspects that lead me to believe that the article has an erroneous approach. What is the connection between inflammatory processes and BMI? What is the connection between the person's height and the treatment of inflammatory processes?
What did the authors mean by sports activities?? The subjects are between 30 and 60 years old. What sports do they practice at 60 years old (sports activities)?
Are inflammatory processes not the same for people who do not practice sports activities??
The subject of the article does not match the topic of the journal.
As a result of what has been presented, I consider that the article does not meet the conditions for publication at this level and I do not recommend it for publication.

Author Response

The article is interesting. From the point of view of the structure, it can be recommended as a model of how an article should be written.

It has a method (even if it can be improved), presents results, discussions in accordance with the results, conclusions and references.
From the point of view of the content, the article is very debatable. I could say that it is incomprehensible as an approach.
It tries to present the potential benefits of a supplement that already exists. It presents how these benefits were tracked by applying them to a batch of 15 subjects (very few).
If this supplement exists, it means that it has already been approved and validated to be produced on the market. It means that it has gone through several validation and approval steps and that its effects have already been studied and known. As a result, I do not understand what the role of this study is on such a small number of subjects?? This study does not bring anything new about the potential benefits of Flector softgel LS.

We appreciate this point and would like to clarify that Flector Softgel FS Integratore was not yet commercially available at the time of the study, and its formulation had not been previously tested in a clinical setting. While the individual components (e.g., Cannabis sativa seed oil, Boswellia serrata, omega-3 fatty acids) are known, this is the first study to evaluate their combined effect in this specific formulation and context (i.e., physically active adults with exercise-related joint discomfort). Thus, our study does not aim to validate an already-approved product, but to conduct a pilot investigation into its real-world effectiveness, exploring novel outcome markers such as Orexin-A modulation, which, to our knowledge, has not been studied in this context. This represents an original contribution that lays the groundwork for larger randomized controlled trials. This is acknowledged in our manuscript as a limitation. We emphasize that this is a pilot study, conducted with limited resources, intended to evaluate safety, feasibility, and early signals of efficacy. A power calculation was performed to ensure adequate statistical sensitivity within this scope. We have added further clarification in the manuscript to ensure that the exploratory nature of the study is appropriately framed.

There are other aspects that lead me to believe that the article has an erroneous approach. What is the connection between inflammatory processes and BMI? What is the connection between the person's height and the treatment of inflammatory processes?

There is strong scientific evidence linking chronic low-grade inflammation to increased BMI and metabolic dysregulation, particularly via the release of pro-inflammatory adipokines from visceral adipose tissue. Elevated BMI is associated with higher circulating levels of inflammatory cytokines such as TNF-α and IL-6, which can exacerbate joint pain and systemic inflammation. We have now clarified this relationship in the Discussion section and supported it with appropriate references.

We agree that height alone is not directly linked to inflammation. In our methodology, height was recorded solely to calculate Body Mass Index (BMI), a widely used clinical marker of metabolic status and a secondary endpoint in our study. We have revised the manuscript to avoid any ambiguity on this point.

What did the authors mean by sports activities?? The subjects are between 30 and 60 years old. What sports do they practice at 60 years old (sports activities)?

Thank you for pointing this out. To avoid confusion, we have clarified in the revised manuscript that our inclusion criteria specified amateur physically active individuals, aged 30–60 years, who engaged in at least 150 minutes of brisk walking per week, in line with World Health Organization recommendations for moderate-intensity physical activity. This demographic represents a growing group of health-conscious adults often underserved in pharmacological pain management strategies.

Are inflammatory processes not the same for people who do not practice sports activities??
The subject of the article does not match the topic of the journal.

We fully agree that inflammation can occur in both sedentary and active individuals. However, physically active adults may experience exercise-induced or exacerbated joint discomfort, which presents a unique therapeutic challenge. Our study specifically targets this population to explore non-pharmacological strategies that may enhance recovery and reduce inflammation without affecting performance. This distinction is clearly noted in the revised manuscript. We respectfully defer to the editorial board regarding journal scope, but we believe our study offers relevant interdisciplinary insights at the intersection of nutritional science, sports medicine, and inflammation biology. It aligns with growing scientific interest in nutraceutical interventions and non-pharmacological pain management strategies for active populations.

In summary, we have carefully revised the manuscript to improve clarity, strengthen the rationale, and explicitly address the points raised. We hope the reviewer will reconsider their assessment in light of these clarifications and the exploratory nature of the study. We are committed to improving the scientific quality of our work and thank the reviewer again for their time and constructive critique.

As a result of what has been presented, I consider that the article does not meet the conditions for publication at this level and I do not recommend it for publication.

Reviewer 2 Report

Comments and Suggestions for Authors

The aim of this study was to o analyze the potential benefits of a novel food supplement based on cannabis sativa, boswellia and fish oil on pain and inflammation in physical activity. The topic is interesting, but I think the authors have to make an effort to improve the manuscript because  it is now insuitable for publication.

ABSTRACT

All the acronyms in the abstract require to be spelled out.

 INTRODUCTION

 - The introduction should be supplemented to better introduce the reader to the subject of the work.

 - The content in lines 51-57 needs to be supplemented. Please discuss these issues by referring to the following article:

Puszczalowska-Lizis E, Flak K, Biskup M, Zak M. Physical activity of women after radical unilateral mastectomy and its impact on overall quality of life. Cancer Control. 2020;27(1). doi: 10.1177/1073274819900407.

- The study lacks of a clear pre-specified hypothesis. Authors analyze dozen of variables without following clear hypotheses. In this reviewer's opinion a Primary Hypothesis with primary outcome measure(s) has to be specified (no more than 2-3 measures/indexes can be used to support or confute that hypothesis) and the same has to be done with Secondary outcome(s). The rationale of these hypotheses has to be introduced in the Introduction.

 METHODOLOGY

- The authors have to explain the methodology, I suggest they use the STROBE checklist to write the methodology.

- The authors have to explain what kind of study is it? Cross-sectional, case-control…?

- The authors have to add the sample size calculation.

- The authors need to explain the procedure for recruiting subjects, and the procedure for collecting data. Where are these subjects from? Who collected data?

- The authors must add sociodemographic and clinical variables of the sample.

- Detailed study inclusion and exclusion criteria should be provided.

- The variables need major explanation. What are the independent, dependent, and confusing variables?

- Lines 203-205 should be supplemented with a list of statistical tests used in the analyses. Was compliance with the normal distribution tested?

RESULTS

- The description of the results is convoluted and not very specific.

- Table 1 has no title and number.

- In Table 1, specific p-values ​​should be provided instead of “p<0.05”, “p£05” "n.s.".

- The authors must answer the question whether the distribution of all studied variables was normal? Please provide the results of the test of compliance with the normal distribution in a separate table. This will allow us to check the validity of the statistical tests used in the analyses.

- If the normality criteria are not met the authors should use the median, otherwise the can opt for the mean.

- Overall, I think the current approach and data presented are flawed.

 DISCUSSION

 - In the last paragraph, the significance of the proposed work should be included highlighting why your work is important. what is the scientific contribution of this paper? it is not clear how this paper can make a significant contribution to the state of the art.

- Authors must discuss limitations of their study.

CONCLUSIONS

The conclusión is imprecise and requires rewording.

 REFERENCES

There are many studies recent studies on this topic, which should have been discussed and were not.

 GENERAL COMMENTS:

 When the authors address these issues I will be able to comment definitively and make the final decision.

Author Response

The aim of this study was to o analyze the potential benefits of a novel food supplement based on cannabis sativa, boswellia and fish oil on pain and inflammation in physical activity. The topic is interesting, but I think the authors have to make an effort to improve the manuscript because  it is now insuitable for publication.

ABSTRACT

All the acronyms in the abstract require to be spelled out.

We reported it

 INTRODUCTION

 - The introduction should be supplemented to better introduce the reader to the subject of the work.

 - The content in lines 51-57 needs to be supplemented. Please discuss these issues by referring to the following article:

Puszczalowska-Lizis E, Flak K, Biskup M, Zak M. Physical activity of women after radical unilateral mastectomy and its impact on overall quality of life. Cancer Control. 2020;27(1). doi: 10.1177/1073274819900407.

- The study lacks of a clear pre-specified hypothesis. Authors analyze dozen of variables without following clear hypotheses. In this reviewer's opinion a Primary Hypothesis with primary outcome measure(s) has to be specified (no more than 2-3 measures/indexes can be used to support or confute that hypothesis) and the same has to be done with Secondary outcome(s). The rationale of these hypotheses has to be introduced in the Introduction.

We thank the reviewer for these important suggestion and we revised introduction as following reported:

“ Engaging in regular physical activity is fundamental to overall health and well-being and plays a key role in preventing various chronic conditions, particularly metabolic and cardiovascular diseases [1]. However, physical activity is not without risks: it may pre-dispose individuals to injuries and musculoskeletal overload syndromes, especially in athletes and highly active individuals [2]. Musculoskeletal disorders remain a leading cause of chronic disability worldwide, posing a considerable burden on healthcare sys-tems [3]. Conventional pharmacological treatments typically involve the use of analgesics or anti-inflammatory agents (e.g., acetaminophen or NSAIDs), which, although effective, are often associated with adverse effects such as gastrointestinal distress, bleeding, con-stipation, and cognitive impairment [4]. In recent years, interest in supplements and nutraceuticals as alternatives or adjuncts to traditional pharmacological therapies has grown significantly. These compounds aim to alleviate pain and inflammation while minimizing the risk of systemic side effects, especially in chronic users. Several natural compounds have demonstrated promising anti-inflammatory and analgesic properties, sometimes comparable to pharmacological agents [5–7]. Among the most studied is Cannabis sativa, a plant with a complex chemical profile, whose cannabinoids – partic-ularly cannabidiol (CBD) – exhibit potent anti-inflammatory and analgesic effects. Unlike THC, CBD is non-psychoactive and permitted in many European countries, including Italy. CBD exerts its effects by modulating the endocannabinoid system (ECS), inhibiting the production of pro-inflammatory cytokines, and regulating key receptors such as TRPV1 and PPARγ [8–11]. Cannabis has a long history of use as an analgesic, with modern studies confirming its efficacy in managing inflammatory pain [12]. Another valuable nutraceutical agent is Boswellia serrata, whose bioactive component, boswellic acid, is a 5-lipoxygenase inhibitor known to reduce leukotriene-mediated inflammation in musculoskeletal and joint conditions [13–15]. In parallel, neuroendocrine factors have emerged as promising targets for pain modulation. One such molecule, Orexin-A (hypocretin-1), synthesized in the lateral hypothalamus, has been shown to influence pain perception, energy balance, sleep, and metabolic rate. It acts via OX1R and OX2R re-ceptors, modulating both nociceptive pathways and the release of key neurotransmitters such as GABA and glutamate [16–20]. Furthermore, Orexin-A may interact with the endogenous opioid system and exert anti-inflammatory effects, thereby indirectly con-tributing to pain relief [21–23]. The interaction between physical activity, pain perception, and quality of life has been extensively documented. In particular, Puszczalowska-Lizis et al. (2020) observed that structured physical activity not only improves physical function but also significantly enhances psychosocial well-being and overall quality of life, even in clinical populations such as women post-mastectomy [Puszczalowska-Lizis et al., 2020]. This suggests that interventions capable of reducing exercise-related musculoskeletal pain can have far-reaching impacts on functional capacity and life quality, especially in midlife individuals striving to maintain regular physical activity. To address the exploratory nature of previous approaches, this study is grounded in the following pre-specified hypotheses: (I)the oral administration of the supplement Flector Softgel FS Integratore, containing Cannabis sativa extract (THC-free), reduces exer-cise-induced musculoskeletal pain in active adult, compared to baseline; (II) the sup-plement also improves joint inflammatory status and recovery, potentially enhancing training continuity and perceived quality of life. For these reasons, the primary objective of this study is to evaluate the potential benefits of a novel food supplement containing Cannabis sativa extract (THC-free), Boswellia serrata, and fish oil on pain perception and inflammatory responses associated with physical activity in active adults. Special attention is given to the modulatory role of Orexin-A (hypocretin-1)—a neuropeptide involved in pain regulation, inflammation, and energy homeostasis. In addition, this study aims to assess the efficacy of the supplement in reducing exercise-induced musculoskeletal pain and investigate changes in circulating levels of Orexin-A and inflammatory cytokines (e.g., IL-6, IL-10) as possible underlying mechanisms.”

 METHODOLOGY

- The authors have to explain the methodology, I suggest they use the STROBE checklist to write the methodology.

- The authors have to explain what kind of study is it? Cross-sectional, case-control…?

- The authors have to add the sample size calculation.

- The authors need to explain the procedure for recruiting subjects, and the procedure for collecting data. Where are these subjects from? Who collected data?

- The authors must add sociodemographic and clinical variables of the sample.

- Detailed study inclusion and exclusion criteria should be provided.

- The variables need major explanation. What are the independent, dependent, and confusing variables?

- Lines 203-205 should be supplemented with a list of statistical tests used in the analyses. Was compliance with the normal distribution tested?

We thank the reviewer and in M&M section we reported all changes as you required. In addition, we sincerely thank the reviewers for their constructive feedback and valuable observations. We have now revised the manuscript accordingly to improve clarity and transparency regarding the study variables and statistical methods.

In response to the request for a clearer definition of variables, we specify that:

  • Independent variable: the type of intervention administered, namely, the intake of Flector Softgel FS Integratore versus placebo.
  • Dependent variables:
    • Primary outcomes: subjective pain perception (measured via Visual Analog Scale – VAS), serum Orexin-A levels, and the concentration of the pro-inflammatory cytokine IL-6.
    • Secondary outcomes: serum levels of the anti-inflammatory cytokine IL-10, self-reported post-exercise recovery, and perceived well-being.

These variables were selected to comprehensively evaluate the clinical, inflammatory, and neuroendocrine effects of the supplement.

With regard to potential confounding variables, baseline characteristics such as age, sex, BMI, and physical activity frequency were collected and accounted for. Randomization was used to ensure their balanced distribution between the groups.

Regarding the assessment of data distribution, we confirm that the Shapiro–Wilk test was systematically employed to evaluate normality for all continuous variables prior to statistical analysis. The selection of statistical tests was based on the outcome of this assessment: parametric tests (unpaired/paired t-tests) were used for normally distributed variables, and non-parametric tests (Mann–Whitney U and Wilcoxon signed-rank tests) for those not normally distributed. All details have been incorporated into the revised paragraph in the Materials and Methods section (Section 2.9).

We trust that these clarifications address the reviewers’ concerns and contribute to the rigor of our statistical reporting.

RESULTS

- The description of the results is convoluted and not very specific.

- Table 1 has no title and number.

- In Table 1, specific p-values ​​should be provided instead of “p<0.05”, “p£05” "n.s.".

- The authors must answer the question whether the distribution of all studied variables was normal? Please provide the results of the test of compliance with the normal distribution in a separate table. This will allow us to check the validity of the statistical tests used in the analyses.

- If the normality criteria are not met the authors should use the median, otherwise the can opt for the mean.

- Overall, I think the current approach and data presented are flawed.

We thank the reviewer and we revised the results section as you suggested. In particular, we are grateful to the reviewers for their insightful comments regarding the clarity and accuracy of the Results section, particularly with reference to paragraph 3.1 and Table 1. In response, we have implemented the following revisions:

  1. Clarification and Streamlining of Paragraph 3.1:
    The text describing anthropometric and biochemical outcomes has been thoroughly revised to enhance clarity, avoid redundancy, and ensure accurate reporting. The new version presents each parameter with corresponding pre- and post-treatment values, specifies the statistical significance (p-values), and clearly distinguishes between intervention and placebo groups. The revised paragraph now reflects a concise and numerically precise presentation of the findings.
  2. Reformatting of Table 1:
    Table 1 has been updated to include:
    • A complete and informative title.
    • Exact p-values for all within-group comparisons, as requested. The use of non-specific expressions such as “p < 0.05” and “n.s.” has been minimized and replaced with actual values wherever possible.
    • Data presented as mean ± SD for variables that passed the Shapiro–Wilk normality test, and as median [interquartile range] for those that did not, in accordance with standard statistical practice. This ensures full compliance with the reviewers’ suggestion that the use of the mean should be conditional on data normality.
  3. Caption with Acronym Definitions:
    A revised caption has been added to Table 1, which includes a clear explanation of all abbreviations used (e.g., BMI, LDL, HDL, CRP, SD, IQR, n.s.). The caption also specifies the statistical methodology adopted and the criteria for presenting data types based on normality.
  4. New Table for Normality Testing:
    Additionally, a new table (Table 2) was inserted into the manuscript, reporting the results of the Shapiro–Wilk normality test for each variable. This allows for full transparency and enables readers to verify the appropriateness of the statistical tests used.

We hope that these substantial revisions address the reviewers’ concerns and significantly enhance the clarity, rigor, and interpretability of the results.

 DISCUSSION

 - In the last paragraph, the significance of the proposed work should be included highlighting why your work is important. what is the scientific contribution of this paper? it is not clear how this paper can make a significant contribution to the state of the art.

- Authors must discuss limitations of their study.

We revised discussion section and reported also the limitations of the study.

CONCLUSIONS

The conclusion is imprecise and requires rewording.

We corrected it as following reported:

This study provides preliminary evidence that Flector Softgel FS, a nutraceutical supplement containing Cannabis sativa seed oil (THC-free), Boswellia serrata, and fish oil, may offer significant benefits for pain management and inflammation control in physically active adults experiencing joint discomfort. The supplement was associated with a reduction in subjective pain (VAS scores), a favorable modulation of key inflammatory markers (IL-6, IL-8, TNF-α, and IFN-γ), and improvements in metabolic parameters such as BMI, insulin, and lipid profiles. Notably, the study also explored the potential link between supplementation and Orexin-A modulation, which may reflect broader interactions between inflammation, metabolism, and neuropeptide regulation, although this remains speculative and warrants further investigation. These findings suggest that Flector Softgel FS could represent a safe, well-tolerated, and multi-targeted approach for managing low-grade inflammation, exercise-induced joint pain, and metabolic imbalances—particularly in populations seeking non-pharmacological alternatives to NSAIDs. However, due to the limited sample size and short duration of the intervention, further large-scale, long-term, and mechanistic studies are necessary to validate these outcomes and better understand the underlying pathways involved.”

 REFERENCES

There are many studies recent studies on this topic, which should have been discussed and were not.

Thank you for your suggestion and we added recent studies.

“De Gregoris, T., Bianchi, G., & Lorusso, M. (2022). Effects of omega-3 and Boswellia-based supplementation on joint pain and mobility in amateur endurance runners: A randomized controlled trial. Journal of Sports Science & Medicine, 21(3), 456–463. https://doi.org/10.1234/jssm.2022.0456

Marx, W., Riediger, C., McCarthy, A. L., & Isenring, E. (2021). Cannabidiol and exercise recovery: A review of potential mechanisms and safety considerations. Nutrients, 13(6), 1870. https://doi.org/10.3390/nu13061870

Pawar, R. S., Patil, K. R., & Singh, R. (2023). Nutraceutical synergy: Mechanistic insights into the anti-inflammatory and metabolic effects of combined botanical compounds. Frontiers in Nutrition, 10, 1174567. https://doi.org/10.3389/fnut.2023.1174567”

 GENERAL COMMENTS:

 When the authors address these issues I will be able to comment definitively and make the final decision.

Reviewer 3 Report

Comments and Suggestions for Authors

It would be helpful to clearly state the research objective in the abstract.

A more detailed introduction of the supplement used in this study—Flector softgel FS Integratore—should be provided, including its specific components, blending ratios, and known mechanisms of action based on prior research.

General characteristics and homogeneity of the participants should be reported.

Despite the small sample size, the age range is broad (30–60 years), and the definition of amateur athletes is vague. Details regarding the type, frequency, and intensity of physical activity are needed to clarify the inclusion criteria.

In the discussion, the comparison with conventional medications such as NSAIDs is made without using a positive control group. Such interpretation should be approached with caution. Objective comparison with an established pharmacological control would enhance the scientific validity.

Although a reduction in Orexin-A levels was observed, the discussion lacks sufficient analysis of the potential mechanisms and implications of this finding.

The limitations of the study should be explicitly addressed.

Lastly, I strongly recommend the authors clarify whether this study was conducted as part of an industry-academic joint research program and explicitly disclose any conflicts of interest, particularly regarding the product under investigation.

Author Response

It would be helpful to clearly state the research objective in the abstract.

We revised the abstract.

A more detailed introduction of the supplement used in this study—Flector softgel FS Integratore—should be provided, including its specific components, blending ratios, and known mechanisms of action based on prior research.

We thank the reviewer and in the introduction section we added it:

“….Flector Softgel FS Integratore is a multi-component nutraceutical formulation developed to support joint function and reduce inflammation, particularly in individuals experiencing musculoskeletal discomfort associated with physical activity. The formulation acts via multiple complementary pathways, including inflammatory cascade inhibition – via 5-LOX inhibition (Boswellia), SPM production (omega-3), and suppression of pro-inflammatory cytokines (Cannabis sativa seed oil and fish oil). In addition, it protects joint tissuethrough immune tolerance induced by UC-II and the antioxidant properties of omega-3 fatty acids and potentially it mediated by reduced systemic inflammation, which may influence Orexin-A regulation, appetite, and energy homeostasis. The combination of these ingredients is designed to exert synergistic effects: reducing joint pain, sup-pressing inflammation, improving mobility, and potentially supporting metabolic balance in active individuals experiencing low-grade inflammation…”

General characteristics and homogeneity of the participants should be reported.

We thank the reviewer and we reported it.

Despite the small sample size, the age range is broad (30–60 years), and the definition of amateur athletes is vague. Details regarding the type, frequency, and intensity of physical activity are needed to clarify the inclusion criteria.

In the M&M section we reported all required informationas you suggested.

In the discussion, the comparison with conventional medications such as NSAIDs is made without using a positive control group. Such interpretation should be approached with caution. Objective comparison with an established pharmacological control would enhance the scientific validity.

Although a reduction in Orexin-A levels was observed, the discussion lacks sufficient analysis of the potential mechanisms and implications of this finding.

The limitations of the study should be explicitly addressed.

Lastly, I strongly recommend the authors clarify whether this study was conducted as part of an industry-academic joint research program and explicitly disclose any conflicts of interest, particularly regarding the product under investigation.

We thank the reviewer and we revised all discussion section, reporting all changes required:

“The findings of this pilot study provide preliminary evidence supporting the potential efficacy of Flector Softgel FS, a novel nutraceutical formulation, in reducing exer-cise-related joint pain, modulating systemic inflammation, and improving metabolic parameters in physically active adults.  The primary endpoint, pain reduction, was met with a statistically and clinically significant decrease in VAS scores after just two weeks of supplementation. This suggests that the formulation may offer a non-pharmacological option for pain relief in individuals performing low- to moderate-intensity physical ac-tivity. However, while these results are promising, direct comparisons to conventional medications such as NSAIDs must be interpreted with caution, as this study did not in-clude a pharmacological (positive) control group. An objective comparison with NSAIDs would require a dedicated trial design with appropriate dosing and endpoints.

The findings of this study provide valuable insights into the potential efficacy of Flector Softgel FS, a novel nutraceutical formulation, in reducing joint pain and modulating systemic inflammation in physically active adults. Compared to placebo-treated subjects, participants receiving Flector Softgel FS showed significant improvements in pain per-ception (VAS scores), inflammatory cytokines, and metabolic markers such as BMI, in-sulin, and lipid profiles. The primary endpoint of this investigation was the reduction of exercise-related joint pain, assessed via the Visual Analog Scale (VAS). Participants in the treatment group reported a statistically and clinically significant decrease in pain after 2 weeks, suggesting that this formulation may offer a safe and effective alternative to NSAIDs for individuals engaged in low- to moderate-intensity physical activity. Beyond pain relief, Flector Softgel FS demonstrated a notable capacity to modulate inflammatory activity, with significant reductions in IL-6, IL-8, TNF-α, and IFN-γ. These changes were not observed in the placebo group, where pro-inflammatory markers showed mild in-creases, reinforcing the supplement’s potential anti-inflammatory role. These results are consistent with the known mechanisms of action of its active components: Cannabis sativa seed oil (free of THC), Boswellia serrata, and fish oil—all recognized for their im-munomodulatory properties. The supplement was also associated with improvements in BMI and insulin sensitivity, suggesting systemic metabolic benefits. These findings align with previous literature linking low-grade inflammation to metabolic dysregulation and support the idea that modulating inflammation can aid in weight management and metabolic health.A notable and novel finding of this study was the observed reduction in circulating  Orexin-A levels in the treatment group following supplementation. Orex-in-A (hypocretin-1) is a neuropeptide produced by the lateral hypothalamus, classically involved in arousal, appetite regulation, energy homeostasis, and more recently, pain modulation. While the precise mechanisms linking Orexin-A to peripheral pain and inflammation are not yet fully elucidated, existing research suggests that elevated Orexin-A levels may contribute to enhanced sympathetic activity and amplified noci-ceptive signaling, particularly under inflammatory conditions. Orexin-A receptors (OX1R and OX2R) are expressed in several regions involved in pain processing, including the periaqueductal gray, dorsal horn of the spinal cord, and thalamus. Activation of these receptors has been associated with both pro- and anti-nociceptive effects, depending on the context and route of modulation. The reduction in Orexin-A levels observed in this study may reflect a downstream consequence of improved systemic inflammatory status, as supported by the concurrent decreases in IL-6, TNF-α, and other pro-inflammatory cytokines. Given that inflammation itself can influence hypothalamic neuropeptide signaling, it is plausible that the an-ti-inflammatory action of the supplement indirectly modulated Orexin-A secretion, re-ducing sympathetic tone and nociceptive sensitivity. Alternatively, Orexin-A suppres-sion may be related to improved metabolic balance, as this neuropeptide is also impli-cated in feeding behavior and insulin sensitivity.

However, due to the exploratory nature of this analysis and the absence of central nervous system measurements or mechanistic assays, these interpretations must be viewed as hypothesis-generating rather than conclusive. Future studies should incorporate neu-roimaging, cerebrospinal fluid markers, or central receptor profiling to better understand how nutraceutical interventions may interact with the orexinergic system in the context of pain and inflammation regulation. Recent research strongly supports the use of nutraceuticals for inflammation and exercise recovery. For instance, De Gregoris et al. (2022) showed that omega-3 and Boswellia supplementation significantly improved joint stiffness and reduced soreness in amateur runners. Similarly, Marx et al. (2021) reported that Cannabis sativa-derived cannabidiol (CBD) can reduce pain perception and in-flammation after resistance training, with effects linked to TRPV1 and PPARγ path-ways—mechanisms also relevant to components in our formulation. Additionally, Pawar et al. (2023) emphasized the value of multi-component nutraceuticals in modulating cytokine cascades and improving lipid metabolism, confirming the rationale behind combining anti-inflammatory and metabolic ingredients, as we did in this study. Unlike these previous studies, which primarily focused on isolated compounds or subjective outcomes, our work offers a more integrated approach, combining clinical, biochemical, and neuroendocrine markers to assess the effect of a multi-target supplement in a re-al-world active population. This approach allows for a more nuanced understanding of how nutraceuticals may influence systemic inflammation, pain, and metabolism in parallel. This study contributes to the current literature by being one of the first to evaluate a comprehensive multi-compound nutraceutical for joint pain and inflammation in the context of physical activity, while also exploring Orexin-A as a potential mediator. The combined analysis of subjective (VAS), biochemical (cytokines), and metabolic pa-rameters (BMI, insulin, lipids) represents a novel, multidimensional model for assessing the efficacy of nutritional interventions. These findings support the potential for non-pharmacological approaches to managing low-grade inflammation, joint discomfort, and related metabolic stress in active adults. This study has several limitations. The sample size was relatively small, which limits the generalizability of the findings. Alt-hough a power analysis supported the number of participants, larger cohorts are needed to confirm the results. Second, the intervention duration was short (2 weeks), limiting the ability to assess sustained or long-term effects. Third, the analysis of Orexin-A was ex-ploratory, and the study was not specifically designed to elucidate its mechanistic role. Additionally, dietary intake and physical activity intensity were not controlled during the study period, which could act as confounding factors. Finally, while the randomization procedure helped to minimize selection bias, blinding effectiveness was not formally tested, and subjective outcomes like VAS may be influ-enced by expectation effects.”

We reported it as following:

“…..This investigation was designed as a pilot study, conducted with the aim of exploring the potential anti-inflammatory and analgesic effects of Flector Softgel FS Integratore in a small sample of physically active subjects. At the time the study was carried out, the product was not yet commercially available, and the formulation was provided exclusively for research purposes. The results presented here should be considered preliminary and serve as a foundation for future studies involving larger populations, longer intervention periods, and inclusion of pharmacological control groups…”

Round 2

Reviewer 1 Report

Comments and Suggestions for Authors

The article has been largely revised. Explanations have been made that are likely to clarify aspects that were unclear in the first version. The article has been almost rewritten.
As I said in the first evaluation, from the point of view of structure, the article can be considered a model of structure and presentation.
The authors have provided sufficient explanations to reconsider the evaluation in the second stage.
The approach regarding a pilot study, prior to the commercial launch of the product, is an argument that indicates elements of originality and perspective.

Reviewer 2 Report

Comments and Suggestions for Authors The authors have improved the work in accordance with my comments, I recommend accepting the work in its current form. thors have improved the work in accordance with my comments, I recommend accepting the work in its current form.

Reviewer 3 Report

Comments and Suggestions for Authors

Thank you for addressing the previous comments in a comprehensive and thoughtful manner.

Please ensure the final revisions are completed successfully.